# Is the malaria short course for program managers, a priority for malaria control effort in Nigeria? Evidence from a qualitative study

IkeOluwapo O. Ajayi[1,2], Olufemi Ajumobi [2,3,4¤]*, Akintayo Ogunwale[5,6], Adefisoye Adewole[2,7], Oluwaseun Temitope Odeyinka[1], Muhammad Shakir Balogun[2,6], Patrick Nguku[2,7], Oluyomi Bamiselu[2], for NFELTP fellows[¶]

1 Department of Epidemiology and Medical Statistics, College of Medicine, University of Ibadan, Ibadan, Nigeria, 2 Nigeria Field Epidemiology and Laboratory Training Program, Federal Capital Territory, Abuja, Nigeria, 3 National Malaria Elimination Program, Federal Ministry of Health, Abuja, Nigeria, 4 School of Community Health Sciences, University of Nevada Reno, Reno, Nevada, United States of America, 5 Department of Health Promotion and Education, Faculty of Public Health, College of Medicine, University of Ibadan, Ibadan, Nigeria, 6 Department of General Studies, Oyo State College of Agriculture and Technology, Igboora, Oyo State, Nigeria, 7 African Field Epidemiology Network Nigeria Country Office, Abuja, Nigeria

¤ Current address: School of Community Health Sciences, University of Nevada Reno, Reno, Nevada, United States of America
¶ NFELTP Fellows (group lead author): Oluyomi Bamiselu, OB, yomzie2003@yahoo.com; Complete membership of the author group can be found in the Acknowledgments
* femiajumobi@gmail.com

**Data Availability Statement:** Yes - all data are fully available without restriction All Key informant interview data files are available from the Zenodo

## Abstract

### Background

A Malaria Short Course (MSC) was conceptualized to build the capacity of program managers for malaria control due to the lack of a single comprehensive broad-based programmatic training in Nigeria. Prior to its implementation, a needs assessment was conducted based on the perspectives of stakeholders to plan and develop the curriculum.

### Methods

This was an exploratory qualitative study. Fifty-six purposively selected stakeholders at local, state and national levels were interviewed. Opinions on the need for training, its perceived impact, priority focus, likelihood of participation, sustainability of and planned support for the MSC were explored using a pretested researcher-designed interview guide. Interviews were audiotape recorded, and the transcripts were subjected to thematic content analysis.

### Results

Participants included Directors of Primary Health Care (50%), State Malaria Program Officers (8.9%), State Directors of Public Health Services (7.1%) and Roll Back Malaria Officers (5.4%). Participants' mean number of years of experience in their current positions was 6.2 (SD 4.7) years. The dominant view was "malaria remains a problem in Nigeria, exacerbated by poor funding, knowledge deficit, lack of training opportunities for program managers and

database. https://zenodo.org/record/3588514#.XpT2VMhKjlU.

**Funding:** This study was supported by Cooperative Agreement Number U2GGH001876 funded by the United States Centers for Disease Control and Prevention through African Field Epidemiology Network (Rebecca Barbriye) to Nigeria Field Epidemiology and Laboratory Training Program. The funders had no role in study design, data collection and analysis, decision to publish, or preparation of the manuscript.

**Competing interests:** The authors have declared that no competing interests exist.

prioritized training budget". A common viewpoint was "to achieve the malaria policy goals, MSC should focus on improving program managers' knowledge of the disease, novel interventions, data audit and use of data for decision making, supportive supervision as well as leadership and management skills. The prioritized thematic areas were malaria epidemiology, case management and data management. The consensus opinion was the MSC would have a positive impact on the performance of program managers. All managerial participants were willing to release their staff for the MSC and encouraged step-down training. However, most participants opined they could not guarantee that their institutions would provide financial support to the MSC attendees.

## Conclusions

Implementing the MSC for program managers was considered essential towards achieving malaria control. Moreover, there is need for prioritized funding and sustainability mechanisms to actualize the implementation of the course.

## Introduction

Nigeria remains the largest contributor to global malaria cases (25%) and deaths (19%) and ranks first among 11 countries targeted for World Health Organization (WHO) High Burden to High Impact (HBHI) approach [1, 2]. The WHO country-led HBHI approach is aimed at putting the global malaria control response back on track after recent resurgence with 217 million and 219 million malaria cases reported globally in 2016 and 2017 respectively [2–4]. Previously there was a decline to 214 million cases in 2015 from 239 million cases in 2010 [5]. While the WHO remains committed to global malaria eradication by 2050 [6], there is an urgent need to strengthen subnational program management capacity in order to achieve this goal. This strategic vision is being championed by African Leaders Malaria Alliance (ALMA) with emphasis on country leadership, governance and accountability [6].

One major driver of the sub-optimal achievement in malaria control is program managers' poor understanding and low level of knowledge of malaria and the related biological, socio cultural and economic drivers of its transmission. The high turnover rate of program managers especially at the state level and lack of training for incoming officers are also major contributors, resulting in the proliferation of newly appointed program managers bereft of technical and programmatic skills for malaria control. Continuous education in form of short courses coupled with carrying out analysis of data generated at state level is deemed an efficient strategy to bridge this gap. The Malaria Short Course (MSC) is carried out over a 3-month period and consists of three workshops. Each workshop runs for 2 to 6 days, separated by periods of field work (two 4-week on-the-job projects). It is tailored after the model used in the Frontline Field Epidemiology Training Program funded by the United States Centers for Disease Control and Prevention and deployed in several countries globally [7]. This is apt for program managers who cannot afford to participate in weeks-long didactic training due to the nature of their jobs.

The United States President's Malaria Initiative-funded Nigeria Field Epidemiology and Laboratory Training Program (NFELTP) is committed to building capacity among malaria control program managers and stakeholders in malaria control, and by extension that of health workers. Thus, the NFELTP planned a short course on malaria to strengthen the capacity of

program managers especially those in malaria control programs, to effectively oversee the malaria control activities in the state and support the implementers to provide quality curative and preventive services while adhering to guidelines. Prior to this study, there was no single comprehensive training course on malaria in Nigeria; rather, trainings were held on thematic malaria areas such as case management, procurement and supply chain management, monitoring and evaluation. Additionally, there is a frequent turnover of trained staff in malaria control at all levels of program implementation and there exists no broad-based training for new program staff. This study was designed as a need assessment from the perspectives of stakeholders for planning and developing curriculum for a comprehensive MSC which seeks to build the capacity of program managers for malaria control in Nigeria. The specific objectives of the study were (1) to assess stakeholders' opinions on the need for a MSC for program managers; (2) to identify priority focus of training and thematic areas that would improve skills of program managers for effective discharge of their duties including supervision skills; (3) to determine the sustainability of, and planned support for MSC, and willingness of stakeholders' organizations to release staff for the MSC

## Methods

### Study setting

Nigeria is the most populous country in Africa with an estimated total population of 203 million for 2019 and ranks 7th in the list of countries by population [8]. It comprises six geopolitical zones, 36 states (plus the Federal Capital Territory of Abuja), and 774 local government areas distributed on a total land area of 91, 770km$^2$. Nigeria, like other tropical countries, has climatic conditions which favour the survival of mosquito vector species and malaria transmission. The duration of the transmission season decreases from year-round transmission in the south to three months or less in the upper north. *Plasmodium falciparum* is the predominant malaria species [9]. The Federal Ministry of Health has a division for malaria control, the National Malaria and Vector Control Division which comprises the National Malaria Elimination Program (NMEP) in Abuja and National Arbovirus Research Centre in Enugu. The NMEP, led by the National Coordinator, coordinates all activities on malaria control in the country in collaboration with State Malaria Program Managers who are responsible for the State Malaria Elimination Programs. A State Malaria Program Manager works collaboratively with the local government area Malaria Focal Persons who coordinate malaria activities at the local government area level. The NMEP collaborates with development partners and other related organizations to plan malaria control activities. This study was conducted by NFELTP in collaboration with NMEP.

### Study design and population

An exploratory qualitative study was conducted among stakeholders working in malaria control programs from April to May 2018. These included NMEP officers, staff of States' Ministries of Health (Directors of Public Health/Disease Control/Primary Health Care, State Malaria Program Managers/Officers), local government area Malaria Focal Persons and Monitoring and Evaluation (M&E) Officers.

### Data collection

The interviews were conducted among stakeholders in malaria control programs in Nigeria that were purposively selected from various health organizations in 10 states, namely: Abia, Akwa-Ibom, Bauchi, Bayelsa, Ebonyi, Ekiti, Kaduna, Kwara, Ogun and Sokoto States as well as

from the Federal Capital Territory Abuja. Participants were selected across the local government area level, State and Federal Ministries of Health and non-governmental organizations for holistic views and representativeness. Overall, 56 key informant interviews were conducted using an open-ended semi-structured interview guide. The sample size was sufficient to reach saturation of views/opinions and key themes. The study participants included NMEP officers, malaria program managers at the local government area and state levels, Directors in the State Ministries of Health, Federal Ministry of Health/National Agencies as well as non-governmental organizations.

A pretested researcher-designed key informant interview guide was used to gather participants' opinions relating to need for a MSC for program managers, how the MSC can help improve the quality of data generated at the local government area and state programs, what the priority focus of training should be that would improve skills of program managers for effective data utilization, what skills are required to supervise program effectively, the thematic areas to consider for the MSC, willingness of organization to release staff for the MSC, sustainability of and planned support for the MSC. The key informant interview guide designed in English language, was pilot tested by three experienced researchers among stakeholders who were not participants in the study (S1 File). The guide was refined accordingly.

Eleven trained male and female research assistants conducted the key informant interviews in English language at the workplace of participants. The research assistants comprised individuals who were medical doctors and field epidemiologists (NFELTP Fellows with at least a Master of Public Health degree) who had previous experience in qualitative research but had no experience in malaria program. Their training comprised acquaintance with the data collection instrument, practical sessions on interviewing skills, note-taking and transcription of recordings. Each interview entailed asking open-ended questions (S1 File) and this was conducted on a one-on-one basis in a place and time of choice of interviewee by a pair of the research assistants. A research assistant served as moderator while another was responsible for note-taking, observing and documenting the interview process and non-verbal cues of an interviewee).

Prior to the commencement of the interviews, the purpose of the study was explained to the interviewees and their voluntary participation was sought. Also, before the interviews, the interviewers did not have any prior personal relationship with the interviewees. Similarly, the interviewers were open-minded and ensured that the interviewees were comfortable during the interviews. All the interviews were audiotape recorded, and notes were taken. Each interview lasted 45–60 minutes. The interviews were held to the point of saturation.

## Data processing and analysis

Data processing commenced with transcription of interview audio recordings immediately after data collection to avoid loss or omission of important details. Eleven NFETLP Fellows performed the transcription while five fellows vetted this (all fellows were involved in the study). The study qualitative data analyst validated the transcribed notes. Thereafter, the transcribed notes were entered into the computer using NVIVO version 12 Pro [10].

Inductive-dominant coding approach was used [11]. Primary codes (parent nodes) and secondary codes (child nodes) were generated based on the content of the data. The codes were linked appropriately to the corresponding quotations. Memos were created as necessary and linked with appropriate codes and quotations. The generated codes and the quotations were reviewed and critiqued carefully by the qualitative data analyst and a team of five experienced qualitative researchers. The group review exercise lasted two days resulting in the removal of repetitive codes and quotations and merging of similar codes and quotations.

Thematic content analysis was performed. Generated themes that facilitated the presentation of the findings were based on the (a) content of the study instruments (b) sample quotes from transcripts; and (c) group reflections (contributions from members of the research team). Following Nowell et al.'s step by step approach to thematic analysis, the verbatim transcript of each interviewee was carefully read and examined theme by theme in comparison with that of other interviewees to identify relevant texts, repeating words, similar phrases and divergent opinions. For each theme, common and peculiar trends as well as similar and divergent opinions were noted. Themes were developed and revised iteratively [12].

Summary of findings were written, and samples of appropriate verbatim quotes were provided. Overall findings of the study were presented using a narrative approach to explore participants' opinions about the need for a MSC for program managers, priority focus of the training, willingness of organizations to release staff and to determine the sustainability of and planned support for the course.

## Ethical considerations

Ethical approval (Ref: UI/EC/18/0089) for the study was obtained from the Institutional Review Board of University of Ibadan/University College Hospital Ibadan, Nigeria. Individual written informed consent was obtained from each of the participants at every stage of the study (none refused participation). There was no risk or harm to participants in this study. Interviews were conducted either in interviewees' personal offices or other convenient places within their work environment where their privacy was guaranteed. Transcribed notes were de-identified. Confidentiality was maintained.

## Results

### Socio-demographic characteristics of participants

The 56 key informant interviewees were professionals who had been involved in malaria control program in their various organizations. Most (96.4%, n = 54) of the key informant interviewees were men. Nearly two-third (62.5%, n = 35) worked with the local government area Primary Health Care (PHC), while 28.4% and 5.4% worked with State and Federal Ministries, respectively. Key informant interviewees who were Directors of Primary Health Care (PHC) had the highest proportion (50.0%), followed by those who were Medical Officers of Health– MOH (10.9%), State Malaria Program Officers (8.9%) and State Directors of Public Health/ Disease Control (7.1%). The mean number of years participants had spent in their present positions was 6.2 (standard deviation: 4.7 years), see Table 1.

### Stakeholders opinion on the need for an MSC for program managers

Participants identified inadequate human resources, knowledge deficit of health staff and lack of training opportunities for health workers as inhibitors of successful malaria programs in Nigeria. Staff attrition arising from abrupt transfer of health workers and retirement of many skilled health workers without replacement accounted for the human resource capacity gap. This lack of skilled health workers resulted in task shifting to community health volunteers to help carry out clinical duties. Participants specified some health workers managing malaria cases at primary health care level and some program officers lacking adequate knowledge about the policy, prevention and treatment guidelines for malaria control programming

**Table 1. Demographic characteristics of participants (N = 56).**

| Characteristics | Frequency |
|---|---|
| **Sex** | |
| Men | 54 |
| Women | 2 |
| **Types of health organization** | |
| Local government area PHC | 35 |
| State Health Ministry | 16 |
| Federal Ministry of Health/NMEP | 3 |
| Non-Governmental Organization | 2 |
| **Designation/present position** | |
| Director of PHC | 28 |
| Medical Officer of Health (MOH) | 6 |
| State Malaria Program Officer | 5 |
| Director of Public Health/Disease Control | 4 |
| Head of Program | 4 |
| Roll Back Malaria Officer | 3 |
| Head of Monitoring and Evaluation (M & E) | 2 |
| Executive Chairman/Secretary of State Health Management Board | 2 |
| Epidemiologist | 1 |
| Chief Executive Director | 1 |
| **Years of experience** | |
| <1 | 3 |
| 1–5 | 30 |
| 6–10 | 10 |
| >10 | 13 |

*Human resource is a problem because even the ones trained are retiring on daily basis; we have staff attrition. And even some staff who are trained might be posted out and the knowledge of new ones coming on board need to be updated. These are the gaps that we need to fill*

(*Participant 5, Director of PHC*).

*The actual implementers of the policy at the grassroots levels who are the health care providers, lack capacity to carry out their assignment effectively and, adequate knowledge on the management of malaria cases*

(*Participant 16, Director of PHC*).

*The biggest gaps we have is that of human resources. There is absolute shortage of human resources especially in my state. The State Government has not been employing for a long period of time. And there is reduction in the number of staff due to death, retirement, transfer and the rest; this has affected implementation of some of the public health programs especially the malaria program. Because of lack of adequate human resource, what we now talk about is task shifting and if you shift task to low level people it becomes a problem*

(*Participant 29, Executive Secretary Primary Health Development Agency*).

Additionally, participants expressed the need for regular malaria training and refresher courses for health workers and program managers to help improve their knowledge of relevant issues in malaria management and control. These will also help in generating quality data that can be used for decision making.

*There is a need for program managers to continually undergo refresher courses so that they can always update their knowledge about the current trends in the management of malaria*

(*Participant 32, MOH*).

## Priority focus of training and thematic areas for inclusion in MSC curriculum

The participants opined that program managers and health workers working on malaria will require diverse additional knowledge and skills that are needed for policy implementation.

*We need the malaria program officers to update their knowledge on goals of national malaria elimination*

(*Participant 21, Director PHC*).

*I think this short course can serve as a refresher course for program managers and the health workers at the lower levels to be able to understand the concept of malaria control and have more understanding on innovations in malaria elimination. It is very important to train those people to have knowledge of basic monitoring and evaluation*

(*Participant 42, M&E officer*).

Other additional knowledge and skills that the participants felt could be harnessed for programmatic implementation include skills on leadership and resources management, knowledge on resource mobilization raising, knowledge of reporting system, skill on advocacy, and Information Communication and Technology (ICT).

*There is a need to enlighten them on leadership and how to manage resources, like management of human resources, materials and others. There is need to build capacity in these areas*

(*Participant 22, Director PHC*).

*Skills in social mobilization, advocacy, skills on planning process and on the development of annual operational plan are needed. Skills on supportive supervision is needed and then skill on job mentoring and the review of data and report writing*

(*Participant 36, Director of Disease Control*).

*There is also need for people to know communication skills, some of the gaps exist when you don't know how to pass an information or how to receive a response, so we need communication skills, then there is need to have a knowledge of how to simplify information to a layman's language and to a native language for all issues including how to translate technical terminologies or issues*

(*Participant 51, Executive Secretary State Primary Health Care Development Agency [SPHCDA]*).

The key informant interviewees prioritized specific areas of focus for training to improve skills of program managers. Data management and data audit skills were recognised as essential to enable data interpretation and appropriate decision making at different administrative levels. Some participants emphasised that the training should place emphasis on data collection, verification and analysis to enhance high quality data reporting. They mentioned that health workers' proficiency in the use of data analysis software such as Statistical Package for Social Sciences (SPSS) and Microsoft Excel should be the major focus of the training. Specifically, they recommended the use of District Health Information System (DHIS), a web-based platform for the National Health Management Information System.

*The management of the data is a great problem, so that is the area of focus I want the organizers of the program to really tailor their effort. The training program should focus on how staff can really be trained on that data management*

(*Participant 23, Director of PHC*).

*The focus should be on data generation, data entry, data analysis, data interpretation and data use using different software to analyze these data*

(*Participant 33, Director of PHC*).

*The training should be on how to be able to cross check records, analyze and be able to use the ICT [information communication technology] system. Every data officer must learn how to access internet and report adequately*

(*Participant 38, MOH*).

*Basically, it is Microsoft Excel and Power Point. Another thing is SPSS*

(*Participant 48, State* Malaria M & E *officer*).

Additionally, other core areas suggested for the training curriculum include data validation, data interpretation and data utilization for better planning

*This course should train officers on data validation, data analysis, interpretation and utilization for better planning*

(*Participant 33, Director of PHC*).

The key informant interviewees opined that the course would improve the quality of data reported at the health facilities and those generated from the local government area, state and national programs. Participants also emphasized the course would improve program managers' knowledge of data management and validation.

*Not everyone is properly trained. If data management is part of the curriculum and they are all trained during the short course program, they will improve on their performance. The method to employ, the strategy to use would have been impacted to them and that will make them do better and give us better report and data on malaria*

(*Participant 31, Director, SPHCDA*)

*The short course on malaria if it is well packaged will result in improved reporting of malaria data for the state using some of the policies and guidelines that will be discussed*

(*Participant 47, Director of Public Health*).

*It is going to really help program managers or other relevant health workers to understand why they generate the data and the quality of the data they generate are important*

(*Participant 41, Integrated Vector Management officer*).

*The course will help program managers to manage data properly and know which information are needed and which ones are not needed. Sometimes when they come for reporting we realize that some information that they give are not necessary and the ones that are necessary are not captured. So, it will improve accurate data collection and collation and documentation by the time this training is conducted*

(*Participant 51, Executive Secretary SPHCDA*).

*We are hoping the short course will be able to build capacity on the concept of M&E and surveillance and also Health Management Information System (HMIS) skills so that they can understand how to use those reports correctly and also how the surveillance system works, the reporting pathway, information flow at each point and that there's also supposed to be feedback*

(*Participant 42, M & E officer*).

Some interviewees mentioned that training should specifically build knowledge of participants on knowledge of epidemiology of malaria and case management of malaria. The interviewees felt that malaria program managers' knowledge of causation, prevention and treatment of malaria and malaria prevention in pregnancy should be improved upon. The participants mentioned that knowledge of health workers in adhering to malaria treatment guideline should be included.

## Sustainability, personnel training and support for the Malaria Short Course

The key informant interviewees were asked questions about the policy on continuing medical education for staff in their establishments. Participants across the various levels (local government area, state and national levels) opined that training and re-training of staff is essential and they mentioned that they have policies that encourage capacity building. Some participants, majorly those working at national level and some working with the State governments mentioned that they usually have annual plans for training of staff. Several participants working in the States Ministries of Health stated that they had policies that allow staff to go for continuing medical education programs. The interviewees working with the local government PHC opined that most of their staff had limited opportunities for training.

Less than one-fifth of the participants mentioned they usually had training funds in the annual budget. While some stated these were the case, the funds were hardly released. Most participants who were local government staff mentioned that funds for training were usually not included in their budgets and thus unable to provide financial support for their personnel who intend to attend the MSC.

*We do have funds for those that are going for in service training. Government is sponsoring them and paying them their salary but for these short courses we do not have budget for that*

(*Participant 16, Director of PHC*)

*No training fund, I mean at the local government area level although at the state we do have trainings. The ones we do at the local government area level is purely our personal sacrifice*

(*Participant 3, Director of PHC*).

All the key informant interviewees were willing to release their staff for the training. Some interviewees, however, mentioned they would not be able to release more than one or two staff at a time for the training due to insufficient number of staff in their workplace. However, some participants mentioned that they would make plans to deploy other staff to assist with the work schedules of the staff going for the training. Most participants mentioned that the short course would have a positive impact on the workers' service delivery.

*I believe the release of my officers will have a positive impact and will promote effectiveness in service delivery so there is need for training and capacity building to promote productivity in the workplace. So, we encourage the learning process and opportunities to promote productivity*

(*Participant 37, Director of Public Health*)

*So long as you are not going to take too many staff at the same time, I guess you will be picking or selecting few staff. It will be impactful. We have to ensure that there are staff on ground to cover for those to attend the course. But in the end if their capacity is built then it will be worth it and impactful*

(*Participant 42, M&E officer*).

## Discussion

In this study, the needs assessment from the perspectives of stakeholders for planning and developing the curriculum for the MSC, revealed that program managers do not have sufficient knowledge for implementing malaria control programs. This was also applicable to health workers managing malaria cases at primary health care level. Nigeria is one of 36 sub-Saharan African countries experiencing a health workforce crisis with a shortage of skilled medical personnel at the primary health care level [13]. Several studies noted that the main challenges to malaria elimination in African countries include insufficient technical capability to inform malaria elimination programs and activities at the national level shortages of experienced human resources [14–16]. Notably, the participants in this study emphasized that without having adequate and well-trained manpower, the goal of eliminating malaria in Nigeria remains far-fetched. This finding underscores the need to tackle malaria prevention and management through appropriate capacity building interventions for program managers.

In this study, various additional knowledge and skills for program managers and health workers working on malaria required for policy implementation, were enumerated. These include knowledge of goals of national malaria elimination, data management, M&E-related and supportive supervision skills. The need for knowledge on malaria policy, malaria case management, reporting system, and how to seek for funds for programs, as well as skills for leadership and resources management, computer and information technology skills were also

identified. These identified skill areas are in line with the internationally accepted competency domains for requisite knowledge, skills, and attitudes for public health practitioners [17]. In the study conducted by Bennibor *et al* among primary health workers in Nigeria, public health science skills followed by communication skills, analytic/assessment skills, and community dimensions skills were the topmost domain skill areas considered to be more important for enhanced performance [13]. These identified implementation skills are in tandem with the recent recommendation in Lancet Commission on malaria eradication which emphasized implementation rather than academic/didactic approach to malaria training [6]. This approach to training which focuses on building sub-national leadership and management capacity among district-level malaria leaders has been piloted in Zimbabwe with initial results of increased productivity, coverage, and quality of malaria program operations, strengthened management and leadership, and improved team performance. Similar results have been reported in Ethiopia [18]. The MSC embodies this strategic and focused approach. The curriculum of the MSC incorporates an implementation approach with period of lectures interspersed by field assignments. To reiterate, the identification of leadership and resource management by participants in the study corroborates the emphasis on practical leadership and management skills as the way forward for malaria eradication [6].

The stakeholders' concerns about lack of adequate knowledge of malaria case management and malaria in pregnancy among health workers support the adduced reasons to focus on building health workers' knowledge of epidemiology of malaria, treatment guidelines, and integrated supportive supervision as well as their management-related skills. Supportive supervision has the potential to improve malaria prevention and case management practices, management of malaria in pregnancy by primary health care workers as well as increase healthcare performance [19].

Training on data management can motivate and empower health workers to recognize the importance of gathering accurate and reliable information. The study participants recommended that the MSC should be designed to focus on improving program managers' skills on data validation and management, as well as improving accurate reporting by health workers, in order to promote the generation of high-quality source data at health care level and the aggregated version at programmatic levels. Nwankwo *et al* revealed in their training intervention study in Kaduna State, Nigeria that training health workers on data management yielded improvement in the quality of data generated at and reported from health centres [20]. Program managers are more likely to be confident in giving strategic direction using high quality data. The need for training of program managers and other health workers on data management to improve quality data and reporting at all levels of implementation cannot be overemphasized.

The suggestion that the MSC should be designed to focus more on data management (data generation, collation, analysis, interpretation and reporting), data auditing and validation and use of relevant software packages including DHIS, HMIS, SPSS and Microsoft Excel is in line with requisite skills that are needed to ensure proper management of data and efficient running of malaria programs. Program managers have important roles to play in interpreting, utilizing and managing data at various health care levels. Improving, their capability to manage and promote quality data for malaria prevention and control will help to reduce the challenges of poor data quality and in the long run, help to attain greater efficiencies in the health system. The MSC curriculum should focus more on these core skills set among others such as epidemiology, malaria case management, social mobilization, and advocacy.

Findings of this study attest to the fact that policies relating to training and re-training of health workers exist at various health care levels. However, it is worrisome that health workers especially those at the local government levels have limited access to training opportunities.

Provision of training opportunities for the different categories of health workers involved in malaria control efforts is crucial for the success of renewed control efforts and for the eventual elimination of malaria [14].

Non-availability of funds for training of health workers constitutes a serious threat to efficient malaria control programs. Notably, at the local government level, training funds are hardly included in annual budgets. It is also unfortunate that health workers at the local government and state levels are faced with situations where budgeted training funds are not being released. This poses a serious challenge to providing appropriate training opportunities for program managers and health workers involved in malaria control and elimination efforts. Governments at various levels need to ensure that sufficient funds for training of health workers are made readily available and are used appropriately to improve the capacity of health workers.

It is quite interesting that stakeholders were generally willing to release and provide some form of support within their capacity to their staff to attend the proposed MSC. This positive action may not be unconnected with their recognition and perception of the positive impact of the training on their staff's performance. A major concern, however, is the persistent uncertainty around provision of financial support for the MSC attendees. This finding underscores the need to intensify efforts on advocacy and development of policies that can guarantee appropriate funding and support for training of health personnel in government health institutions. There may also be a need to promote private-public partnership in the provision of appropriate training programs for health workers. It is the desire of WHO that every country focused on malaria elimination or control should involve and collaborate effectively with competent non-governmental partners for training-related support [21]. Hence, concerted efforts need to be put in place to explore feasible private-public partnership models and strategies that can help to develop a sustainable malaria training program for health workers and malaria program managers in Nigeria.

This study had limitations. It was a cross-sectional qualitative study and thus the findings are not generalizable [22]. Qualitative research focuses on description of opinions or experiences. Moreover, the findings of this study may exhibit transferability to similar scenarios which might involve development of training curriculum fit for purpose. Additionally, it offers in-depth insight in consideration for malaria training curriculum.

The first cohort of participants has been trained on the MSC and this NFELTP-supported course presents a singular opportunity for comprehensive broad-based training of malaria program managers in Nigeria (S2 File).

## Conclusions

Implementing the MSC for program managers was considered essential towards achieving malaria elimination. It was specifically noted the MSC would help in reducing challenges facing program managers at various levels of implementation as well as provide opportunities for generating high quality data that can be used for decision making. Data management, validation and utilization skills for better planning, and knowledge of malaria epidemiology were some of the priority focus areas suggested for the MSC. Stakeholders were willing to release their staff for the MSC and provide moral support but financial sponsorship for staff to attend the course could not be guaranteed. The continual implementation of the MSC requires the need for sponsorship and sustainability mechanisms to actualize the organizations' willingness to release staff and achieve the goals of training. Governments at various levels need to ensure sufficient funds for training of health workers on the MSC are readily available. Concerted efforts need to be put in place to explore feasible private-public partnership models and

strategies that can help to develop a sustainable malaria training program for malaria program managers in Nigeria.

## Supporting information

**S1 File. Key informant interview guide.**
(PDF)

**S2 File.**
(PDF)

**S1 Dataset. Key informant interview data.** https://zenodo.org/record/3588514#. XpT2VMhKjIU.
(TXT)

## Acknowledgments

The authors are grateful to National Malaria Elimination Program Nigeria and the Directors of Public Health, Directors of Primary Health Care and Disease Control, Malaria Program Managers and local government area Primary Health Care Coordinators of Abia, Akwa-Ibom, Bauchi, Bayelsa, Ebonyi, Ekiti, Kaduna, Kwara, Ogun and Sokoto States and Federal Capital Territory Nigeria, for their immense support and contribution towards the success of this study. Additionally, the roles of Mark Maire and Laura Steinhardt (United States Centers for Disease Control and Prevention) in the development of the curriculum are highly appreciated. The findings of this study were presented, and feedback received at the 2018 Nigeria Centre for Disease Control /Nigeria Field Epidemiology and Laboratory Training Program Annual Scientific Conference which held in Abuja, Nigeria from 4–6 September 2018.

NFELTP fellows: Oluyomi. Bamiselu (yomzie2003@yahoo.com), Bountain Tebeda, Ntiense. Umoette, Joseph Agboeze, Godwin Okezue, Bosede Alowooye, Istifanus Waziri, Joshua Difa, Taiwo Olasoju, Hannatu Dimas, Ismaila Ibrahim, Abubakar Danmafara, Biobelu Abaye, Tamuno-Wari Numbere, Chindima Amuzie, Nwenyi Okoro, Pius Ononigwe, Chindima Emma-Ukaegbu, Olusola Hassan Ajayi, John Ojo, Hakeem Yusuf, Olukorede Ifedolapo Ikwunne, Jibreel Omar Muhammad, Salisu Isah, Oluseyi Akano, Jenom Danjuma, Nsisong Asanga, Augustine Dada, Eric Edrah, Amina Umar, Adaora Eneja, Irene Esu.

## Author Contributions

**Conceptualization:** IkeOluwapo O. Ajayi, Olufemi Ajumobi.

**Data curation:** IkeOluwapo O. Ajayi, Olufemi Ajumobi, Akintayo Ogunwale, Oluwaseun Temitope Odeyinka.

**Formal analysis:** IkeOluwapo O. Ajayi, Olufemi Ajumobi, Akintayo Ogunwale, Oluwaseun Temitope Odeyinka.

**Funding acquisition:** Patrick Nguku.

**Investigation:** IkeOluwapo O. Ajayi, Olufemi Ajumobi, Akintayo Ogunwale, Adefisoye Adewole, Oluwaseun Temitope Odeyinka, Muhammad Shakir Balogun, Patrick Nguku, Oluyomi Bamiselu.

**Methodology:** IkeOluwapo O. Ajayi, Olufemi Ajumobi, Akintayo Ogunwale, Oluwaseun Temitope Odeyinka.

**Project administration:** IkeOluwapo O. Ajayi, Olufemi Ajumobi, Adefisoye Adewole, Oluwaseun Temitope Odeyinka, Muhammad Shakir Balogun, Patrick Nguku.

**Resources:** Muhammad Shakir Balogun, Patrick Nguku.

**Software:** Akintayo Ogunwale.

**Supervision:** IkeOluwapo O. Ajayi, Olufemi Ajumobi, Akintayo Ogunwale, Oluwaseun Temitope Odeyinka, Muhammad Shakir Balogun.

**Validation:** IkeOluwapo O. Ajayi, Olufemi Ajumobi, Akintayo Ogunwale, Oluwaseun Temitope Odeyinka.

**Visualization:** IkeOluwapo O. Ajayi, Olufemi Ajumobi, Akintayo Ogunwale, Oluwaseun Temitope Odeyinka.

**Writing – original draft:** IkeOluwapo O. Ajayi, Olufemi Ajumobi, Akintayo Ogunwale.

**Writing – review & editing:** IkeOluwapo O. Ajayi, Olufemi Ajumobi, Akintayo Ogunwale, Adefisoye Adewole, Oluwaseun Temitope Odeyinka, Muhammad Shakir Balogun, Patrick Nguku.

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
