## [Decision Letter · Decision Letter 0]

13 Feb 2020

PONE-D-19-35639

Is malaria short course for program managers, a priority for malaria control effort in Nigeria? Evidence from a qualitative study

PLOS ONE

Dear Dr Ajumobi,

Thank you for submitting your manuscript to PLOS ONE. After careful consideration, we feel that it has merit but does not fully meet PLOS ONE’s publication criteria as it currently stands. Therefore, we invite you to submit a revised version of the manuscript that addresses the points raised during the review process.

We would appreciate receiving your revised manuscript by Mar 29 2020 11:59PM. To enhance the reproducibility of your results, we recommend that if applicable you deposit your laboratory protocols in protocols.io, where a protocol can be assigned its own identifier (DOI) such that it can be cited independently in the future. For instructions see: http://journals.plos.org/plosone/s/submission-guidelines#loc-laboratory-protocols

We look forward to receiving your revised manuscript.

Kind regards,

Masamine Jimba

Academic Editor

PLOS ONE

Additional Editor Comments (if provided):

P4: Abstract/Conclusion-It is better to cut “The first cohot of….trained on MSC.” This sentence is not related with the results.P7: “Materials and methods” should be “Methods”P8: and other pages: Please avoid abbreviations which are not familiar with international readers: LGA, NMEP, NAVRC, SMEPs, etc. There are many others.P9: Please add a short description of the malaria short course as an appendix.P13L2: Please be more specific about under-qualified ones. Are they community health volunteers?P21: “All the key informants” mean 56 professionals? Stakeholders, professionals, key informants…these terms are interchangeably used?P22: It is better to show a summary of major findings in the first paragraph of discussion section, with no reference, no discussion. From the second paragraph, discussion of each major finding may start.

Journal Requirements:

2. Please address the following:

- Please include additional information regarding the interview guide used in the study and ensure that you have provided sufficient details that others could replicate the analyses. For instance, if you developed a guide as part of this study and it is not under a copyright more restrictive than CC-BY, please include a copy, in both the original language and English, as Supporting Information. In addition, please provide details of any pre-testing of this guide that took place.

- Please ensure you have thoroughly discussed any potential limitations of this study within the Discussion section, for example the potential bias introduced when using qualitative data and the skewed nature of the dataset.

3. During your revisions, please confirm whether the wording in the title is correct and update it in the manuscript file and online submission information if needed. Specifically, ensure that the title is grammatically correct.

5. One of the noted authors is a group or consortium: NFELTP fellows. In addition to naming the author group, please list the individual authors and affiliations within this group in the acknowledgments section of your manuscript. Please also indicate clearly a lead author for this group along with a contact email address.

6. We note you have included a table to which you do not refer in the text of your manuscript. Please ensure that you refer to Table 1 in your text; if accepted, production will need this reference to link the reader to the Table.

Reviewers' comments:

Reviewer's Responses to Questions

**Comments to the Author**

1. Is the manuscript technically sound, and do the data support the conclusions?

Reviewer #1: Partly

Reviewer #2: Partly

Reviewer #3: Partly

2. Has the statistical analysis been performed appropriately and rigorously? 

Reviewer #1: N/A

Reviewer #2: Yes

Reviewer #3: N/A

3. Have the authors made all data underlying the findings in their manuscript fully available?

Reviewer #1: No

Reviewer #2: Yes

Reviewer #3: Yes

4. Is the manuscript presented in an intelligible fashion and written in standard English?

Reviewer #1: No

Reviewer #2: Yes

Reviewer #3: No

5. Review Comments to the Author

Reviewer #1: This study addresses a very important issue in malaria elimination in Nigeria which is the capacity building of frontline health workers. However, this paper is not properly structured and a number of major issues must be addressed.

1) The authors did not include any checklists for qualitative research such as COREQ or SRQR. Please do so.

2) A more detailed sampling strategy and inclusion/exclusion criteria must be included. For example, under the “Data Collection” subheading, the authors mentioned that “the interviews were conducted among 60 purposively selected stakeholders”. However, in the results, the authors reported “the participants were 56 professionals”. What happened to the other 4 stakeholders?

3) In “Data processing and analysis”, in what language were the interviews conducted?

4) The presentation of Table 1 could be improved.

5) In the discussion, the authors did not discuss any limitations or the potential sources of bias at all. Please discuss the limitations and the potential sources of bias as this will strengthen the scientific value of the paper.

6) The title is a little confusing. The title only mentioned “program managers” but in the discussion and conclusion, the authors also mentioned “relevant health workers”.

Minor issues

7) Please include line numbers in the manuscript for easier review.

8) The authors used “6.2 ± 4.7 years”. Is this an appropriate way to present standard deviation?

9) I suggest that the whole manuscript be sent for scientific editing. For example, there are many minor typo errors.

In Page 5, line 3, “However, there 217 and 219 million malaria cases…”

Page 5, line 13, “…certified to be malaria-free by WHO [5]).”

Page 6, the authors used “%” and “percent” interchangeably. Please be consistent.

Please check the reference for typo errors. For example, reference number 7, “World Healthj Organization…”

Reviewer #2: Reviewer’s comments:

The authors aimed to examine the post training opinions of the Malaria short course for program managers. This qualitative study was done on 60 key interviewees, and involved key informant interviews. The manuscript is well written and detailed results “statements” given. However, I have several comments that may improve this manuscript further.

Major comments:

The methods are not clear in a number of areas.

1. Study design is not mentioned. In the section responsible for this, it was mentioned “an explorative study”. Explorative studies do not specify study design.

2. The analysis plan lack theoretical framework for the qualitative analysis. It would introduce and describe the theory that explains the “why” in the research problem.

3. Both introduction and discussion are general. There is lack of focus in the specific study and context thereof. These sections need to be focused into malaria in the context of the leaders giving the strategic direction/leadership in malaria program, rather than the burden and epidemiology thereof. Only the last two paragraphs speak of this.

4. The conclusion is somehow inflated. I have gone through the evidence presented, and could not see where the PM were reluctant to provide funding/scholarships for their employees’ trainings.

Minor comments:

Abstract

1. Not clear if the study was done after the training or before the training

2. Methods:

Study design is missing

Sampling missing (even the size is missing) though not common for qualitative study, but at least observation to the “Principal of saturation” is essential

Development of tools and content thereof

Analysis plan and theory involved

3. Conclusion: issues with payment for training as explained above

4. Recommendation is missing

Background:

5. Too broad and not focused to the problem at hand. The focus needs to be narrowed to high level workers (Program managers) and rationale for this.

6. Clear problem is missing and implication thereof

7. Clear research questions are missing and therefore SMART objectives

Methods:

8. Study design missing

9. Study area: the focus needs to be contextualized for this study. Focus on programs in Nigeria, leadership, influence they have on policy and implementation etc.

10. Data collection needs to be expanded to also include how the questions were asked, preps of the RAs (if used), training, pre-test (where, how), and people involved in interview, notes taking etc.

11. Re-write analysis plan with necessary contents. Including the theoretical framework, steps in analysis etc.

12. Re-write selection (convenience) and reasons for selection of interviewee

13. Explain about the tool, development, and content thereof

14. Ethical consideration needs to be

Results:

15. Well written, but miss explanation for each result displayed. You can reduce the quotations if they repeat

Discussion:

16. focus on the context.

Conclusion:

17. Conclusion on payment/support has no results attached

18. Add recommendations

Reviewer #3: Authors of this article conducted an exploratory questionnaire survey among stakeholders and program managers/officers who were working on malaria control programs in Nigeria from April to May 2018, and evaluated the importance of “Malaria Short Course” to build capacity of stakeholders and program managers/officers for malaria control and elimination in the country. This is a relevant study to improve a capacity among stakeholders and program managers/officers to achieve the goal of malaria control and elimination in the country.

Major comments

1. It is not clear a definition of “short” or “long” for “Malaria Short Course”. It is better to clarify the definition of “short” in this context.

2. Expressions, especially in the Results section, seemed redundancy. I recommend the authors to revise the texts to avoid redundancy and make them concise. Some description in the results and discussions were also partially overlapped.

3. It is better to summarize comments and challenges obtained from the study participants in a table.

Minor comments

1. Grammatical errors, such as, singular ＆ plural should be corrected.

2. The authors described “Malaria Short Course” and “malaria short course”. This should be uniformed.

3. In Page 15, Comment of a Participant 42, M&E officer

“HMIS” should be spelled out “Health Management Information System (HMIS)” since this word appeared first time.

4. In Page 19, Comment of a Participant 48, State Malaria M&E officer

Please edit “Microsoft excel and power-point” to “Microsoft Excel and Power Point”.

6. PLOS authors have the option to publish the peer review history of their article (what does this mean?). If published, this will include your full peer review and any attached files.

Reviewer #1: No

Reviewer #2: No

Reviewer #3: No

---

## [Author Response · Author response to Decision Letter 0]

4 Jun 2020

The response to reviewers and editors have been uploaded in a file tagged "Response to reviewers May, 2020" in the manuscript submission system.

---

## [Decision Letter · Decision Letter 1]

3 Jul 2020

PONE-D-19-35639R1

Is the malaria short course for program managers, a priority for malaria control effort in Nigeria? Evidence from a qualitative study

PLOS ONE

Dear Dr. Olufemi Ajumobi,

Thank you for submitting your manuscript to PLOS ONE. After careful consideration, we feel that it has merit but does not fully meet PLOS ONE’s publication criteria as it currently stands. Therefore, we invite you to submit a revised version of the manuscript that addresses the points raised during the review process.

We look forward to receiving your revised manuscript.

Kind regards,

Masamine Jimba

Academic Editor

PLOS ONE

Additional Editor Comments (if provided):

The revised manuscript has been greatly improved. However, as the reviewers mentioned, more careful revision is necessary to improve its quality. For any qualitative study, readability is crucial if the authors want readers to read it through. Native English speakers’ editing may not be sufficient unless they are well experienced in academic English writing. Below are additional comments.

1. P3 L18: Please do not use ± in academic writing. 6.2 (SD 4.7) is better.

See below site:

https://www.ncbi.nlm.nih.gov/pmc/articles/PMC2959222/#CIT2

2. P4 L13: Are these all MeSH key words? It is better to use MeSH keywords as much as possible to be quoted in the future.

3. P6 L20: LGA is used in here as abbreviation, and then in P7 L9. As LGA is not a common abbreviation for international leaders, it is better to spell it out always.

4. P6 L18-P7 L2: Five objectives are written in here. In the results, some of these five objectives are not well addressed as major objectives. It is better to modify either these objectives or results (subheadings), so that the objectives and results can be well matched.

5. P8 L2: FCT should be always spelled out.

6. P8 L14, L9: Key informant interview is abbreviated as KII twice. As KII is not frequently used, always spell it out.

7. P9 L6 and many other places: Please use male and female only as adjectives. It is better to use men and women. Male assistants are fine.

8. P10: One paragraph is too long. One paragraph should have only one message. L14 may be a place to start a new paragraph. Please check other long paragraphs, too.

9. P12 Table 1. As the number is only 56, percentage may not be necessary in this table.

10. P18 L20: MSC is forgotten.

11. P22 L22: Usually qualitative research is not conducted for generalization. Is this statement based on some current books or academic papers? If so, please quote it.

12. P26~: Reference section is not professionally done. Sometimes journal names are abbreviated, sometimes they are fully spelled out.

Reviewers' comments:

Reviewer's Responses to Questions

**Comments to the Author**

1. If the authors have adequately addressed your comments raised in a previous round of review and you feel that this manuscript is now acceptable for publication, you may indicate that here to bypass the “Comments to the Author” section, enter your conflict of interest statement in the “Confidential to Editor” section, and submit your "Accept" recommendation.

Reviewer #1: (No Response)

Reviewer #2: All comments have been addressed

Reviewer #3: All comments have been addressed

2. Is the manuscript technically sound, and do the data support the conclusions?

Reviewer #1: Yes

Reviewer #2: Yes

Reviewer #3: Yes

3. Has the statistical analysis been performed appropriately and rigorously? 

Reviewer #1: N/A

Reviewer #2: Yes

Reviewer #3: N/A

4. Have the authors made all data underlying the findings in their manuscript fully available?

Reviewer #1: Yes

Reviewer #2: Yes

Reviewer #3: Yes

5. Is the manuscript presented in an intelligible fashion and written in standard English?

Reviewer #1: Yes

Reviewer #2: Yes

Reviewer #3: Yes

6. Review Comments to the Author

Reviewer #1: Minor comments:

Overall, the draft is getting better and easier to read. I would suggest one final proofreading for minor errors.

1) Page 5 Line 4: high Impact -> High Impact? (Capitalize the ‘H’?)

2) Page 6 Line 4: The United President’s -> The United States President’s?

3) COREQ Checklist was not submitted as an attachment?

4) Page 10 Line 5: Please check the grammar of this sentence: “further validated by the study experienced qualitative data”

5) Page 10 Lines 16-17: Grammar: “Following Nowell et al’s step by step approach to thematic analysis.” Not a complete sentence.

6) Page 22 Lines 14-20: Private-public partnership is important and recommended by the WHO. If possible, include a citation and a one-sentence description of the example of the effectiveness of private-public partnership in other similar settings, if there is any.

7) Page 22 Line 24: “Generalizability is not the goal” -> This sentence is redundant.

Reviewer #2: The reviewers have addressed all my comments adequately. The manuscript can now be accepted for publication

Reviewer #3: The manuscript was improved after the revision. However, some minor errors should be edited.

In the introduction, the authors mentioned that “The term “short course” entails a training delivered for short duration (four to five days) interspersed by period of practicum/field assignments,”…. However, Supporting Information S2 shows that “Malaria short course is carried out over a 3-month period”. The workshop is 2 to 6 days, but on-the-job project is 4 weeks. Therefore, the description in the Introduction should be edited.

I suggest to the authors that the manuscript should be checked by an English editor because there are still many grammatical errors. For example, article (the) is missing in the key words, such as “MSC”, “LGA”.

Other minors

Description should be consistency.

In page 6, line 16 in Introduction: Please correct “malaria course short (MSC)” to “malaria short course (MSC)”.

In page 18, line 20 in Discussion: Please correct “the short malaria course” to “the MSC”.

In page 9, line 13 in Methods: Please correct “the researcher assistants” to “the research assistants”.

Only once would be enough to describe full spell with abbreviation.

In page 6, line 18 in Introduction: Please correct “malaria short course (MSC)” to “MSC”.

In page 7, line 8 in Methods: Please correct “Local Government Areas (LGAs)” to LGAs.

In page 18, Line 25 in Discussion: Please correct “Africa countries” to “African countries”.

In page 19, line 11-12 in Discussion: Please correct “computer and information technology skills and were also identified.” to “computer and information technology skills were also identified.”.

7. PLOS authors have the option to publish the peer review history of their article (what does this mean?). If published, this will include your full peer review and any attached files.

Reviewer #1: No

Reviewer #2: **Yes: **Bruno Fokas Sunguya

Reviewer #3: No

---

## [Author Response · Author response to Decision Letter 1]

5 Jul 2020

The response to reviewers and editors have been uploaded in a file tagged "Response to reviewers" in the manuscript submission system.

---

## [Editor Report · Decision Letter 2]

10 Jul 2020

Is the malaria short course for program managers, a priority for malaria control effort in Nigeria? Evidence from a qualitative study

PONE-D-19-35639R2

Dear Dr. Olufemi Ajumobi,

We’re pleased to inform you that your manuscript has been judged scientifically suitable for publication and will be formally accepted for publication once it meets all outstanding technical requirements.

Kind regards,

Masamine Jimba

Academic Editor

PLOS ONE

Additional Editor Comments (optional):

This time, the manuscript has been well revised. Please make a few more minor revisions.

1 Clean text page 10 L4 and L11: (AO) is not necessary.

2 Clean text page 17 L22, 23: Like other places in the main text, capitalization is not necessary for local government area.
---

## [Editor Report · Acceptance letter]

14 Jul 2020

PONE-D-19-35639R2 

Is the malaria short course for program managers, a priority for malaria control effort in Nigeria? Evidence from a qualitative study 

Dear Dr. Ajumobi:

I'm pleased to inform you that your manuscript has been deemed suitable for publication in PLOS ONE. Congratulations! Your manuscript is now with our production department. 

Kind regards, 

on behalf of

Professor Masamine Jimba 

Academic Editor

PLOS ONE